# COVID-19 Pathology Sheds Further Light on Balance between Neutrophil Proteases and Their Inhibitors

**DOI:** 10.3390/biom13010082

**Published:** 2022-12-30

**Authors:** Vasuki Silva, Marko Radic

**Affiliations:** Department of Microbiology, Immunology and Biochemistry, College of Medicine, University of Tennessee Health Science Center, Memphis, TN 38163, USA

**Keywords:** alpha-1-antitrypsin (AAT), neutrophil elastase (NE), NETosis, PAD4, ARDS, COPD, neutrophils, COVID-19, TMPRSS2

## Abstract

Excessive neutrophil influx and activation in lungs during infections, such as manifest during the ongoing SARS CoV-2 pandemic, have brought neutrophil extracellular traps (NETs) and the concomitant release of granule contents that damage surrounding tissues into sharp focus. Neutrophil proteases, which are known to participate in NET release, also enable the binding of the viral spike protein to cellular receptors and assist in the spread of infection. Blood and tissue fluids normally also contain liver-derived protease inhibitors that balance the activity of proteases. Interestingly, neutrophils themselves also express the protease inhibitor alpha-1-antitrypsin (AAT), the product of the SERPINA-1 gene, and store it in neutrophil cytoplasmic granules. The absence of AAT or mutations in the SERPINA-1 gene promotes lung remodeling and fibrosis in diseases such as chronic obstructive pulmonary disease (COPD), and acute respiratory distress syndrome (ARDS) and increases the risk of allergic responses. Recent observations point to the fact that reduced activity of AAT presents a major susceptibility factor for severe COVID-19. Here, we focus attention on the mechanism of neutrophil elastase (NE) in NET release and its inhibition by AAT as an additional factor that may determine the severity of COVID-19.

## 1. Introduction

Neutrophils play a central role in innate immunity as first-line defenders against invading pathogens and in response to injury. In infection or injury, neutrophils rapidly migrate from the peripheral blood into the tissues and follow a chemotactic gradient to the site of inflammation [1,2,3]. They execute different host defense mechanisms upon encounter with pathogens, which include phagocytosis, the release of reactive oxygen species (ROS), and degranulation [4]. Activated neutrophils secrete proteolytic enzymes, notably, neutrophil elastase (NE), most commonly during the release of neutrophil extracellular traps (NETs). NETs consist of chromatin fibers that are thought to limit the spread of infection [5,6]. NE associates with NETs and may promote NET formation [7,8]. Degranulation, the release of granule contents, can be directed toward phagocyted (intracellular) pathogens or toward extracellular pathogens. Cytoplasmic granules contain proteases, and their release from the cells may serve to disassemble extracellular matrix components (ECM), such as during the formation of migratory channels that facilitate neutrophil extravasation [9]. Though these mechanisms are vital to combat infection, excessive accumulation or inappropriate activation of neutrophils can cause host tissue damage. Thus, neutrophils participate in the pathogenesis of a range of acute and chronic lung diseases, including emphysema, acute lung injury (ALI), chronic obstructive pulmonary disease (COPD), bronchiectasis, and cystic fibrosis. Additionally, neutrophils exacerbate disease severity in lung infections caused by influenza and SARS-CoV-2 [10,11]. Therefore, neutrophil activity must be tightly regulated to prevent the excessive release of proteolytic enzymes and pro-inflammatory cytokines that could exacerbate inflammation. NE is one of the most potent neutrophil serine proteases and it is stored in cytoplasmic granules. Though NE has a pivotal role in fighting pathogens, unchecked NE itself can contribute to disease manifestations. Notably, excessive neutrophil influx and activation in lungs during infections have brought NETs and the concomitant release of neutrophil proteases that damage surrounding tissues into sharp focus. NE inflicts direct damage to ECM by the digestion of structural components including elastin and collagen [10]. NE proteolytic activity can also be tied to increased mucus production [12], impaired clearance through cilia, and degradation of surfactant proteins [13], ultimately leading to airway epithelial damage and dysfunction [14]. COPD is one example, where NE-driven tissue damage plays an important role in pathogenesis and disease progression [15]. Conversely, blood and tissue fluids contain liver-derived protease inhibitors that regulate the activity of neutrophil proteases. Alpha-1-antitrypsin (AAT) is a prominent antiprotease that inhibits the activity of NE by forming a covalent complex that blocks further proteolytic activity. 

AAT is a glycoprotein that is mainly produced by the liver and released to the circulation in high concentrations (1.5–3.5 g/L) in a healthy individual. AAT concentration may increase several folds during injury and acute inflammation which may be required to regulate the activity of NE and prevent collateral tissue damage [15]. When NE is secreted, its activity is limited to the vicinity of the inflamed area because abundant AAT forms NE-AAT complexes within a sphere, whose radius is determined by the relative concentration of AAT (Figure 1A). This prevents the dispersal of active NE beyond its immediate surroundings and limits potential off-target effects of NE at more distant sites. As NE diffuses into an excess of its inhibitor, its activity is confined to a location very near to the site of its release. Therefore, the balance between AAT and NE is vital to preserving health in tissues, such as the lung. Should this balance be disturbed, the affected individual becomes more susceptible to diseases such as emphysema, ALI and COPD. 

Genetic AAT deficiency (AATD) is a disorder that arises due to mutations in the SERPINA-1 gene, which encodes for AAT. The most harmful phenotype of AATD is caused by the ZZ mutation. Insufficient AAT production due to the ZZ genotype lowers the serum AAT level drastically, allowing NE activity to spread and impact a wider area, including sites more distal to the initial inflammatory insult (Figure 1B). AATD was identified as a potential risk factor in lung diseases that may grow progressively worse with age and be aggravated by environmental triggers such as smoking. Importantly, AATD has been proposed to affect the disease severity of the SARS-CoV-2 outbreak, which has caused several million deaths worldwide. According to emerging data, as many as 33% of AATD patients require hospitalization or critical care after infection with SARS-CoV-2. One possible reason for this outcome is that AAT inactivates transmembrane protease serine 2 (TMPRSS2), which mediates the priming of viral spike protein and facilitates virus entry [16]. Recent evidence suggests NE may also be hijacked by SARS CoV-2 to prime the S1/S2 interface and promote infection [17]. Hence, Individuals with insufficient functional AAT, therefore, may not receive the antiviral effects of AAT and face an increased risk of succumbing to severe disease. 

Both AAT and NE affect the pathogenesis of COVID-19. A substantial loss of AAT was observed in the airways of COVID-19 patients with acute respiratory distress syndrome (ARDS) complications [18], which may favor NET release and NE dispersal. NETs are cytotoxic to endothelial and epithelial cells [19] and may contribute to lung pathology. NETs also induce platelet activation, aggregation, and thrombosis, thus affecting blood flow in lung capillaries and small arteries [17]. NE embedded in NETs may also facilitate the extravasation of additional waves of neutrophils by degrading heparan sulfate proteoglycans, an essential component of lung parenchyma [20]. In sum, AAT-mediated inhibition of NE activity may ameliorate fatalities in COVID-19. Several clinical trials are ongoing to provide AAT supplementation in COVID-19 patients. Below, we present detailed views of NE and AAT, outline their interactions, and highlight potential consequences of an imbalance between these adversarial proteins.
Figure 1Alpha-1 Antitrypsin (AAT) inhibition of neutrophils elastase (NE). (**a**) AAT (Blue) is abundant and uniformly distributed in circulation. AAT constrains the activity of NE (red) by forming a 1:1 AAT/NE complex (AAT; green, NE; yellow). (**b**) Relatively low concentration of AAT due to inherited AAT deficiency leaves large proportion of NE unopposed, allowing NE to spread into a larger area.
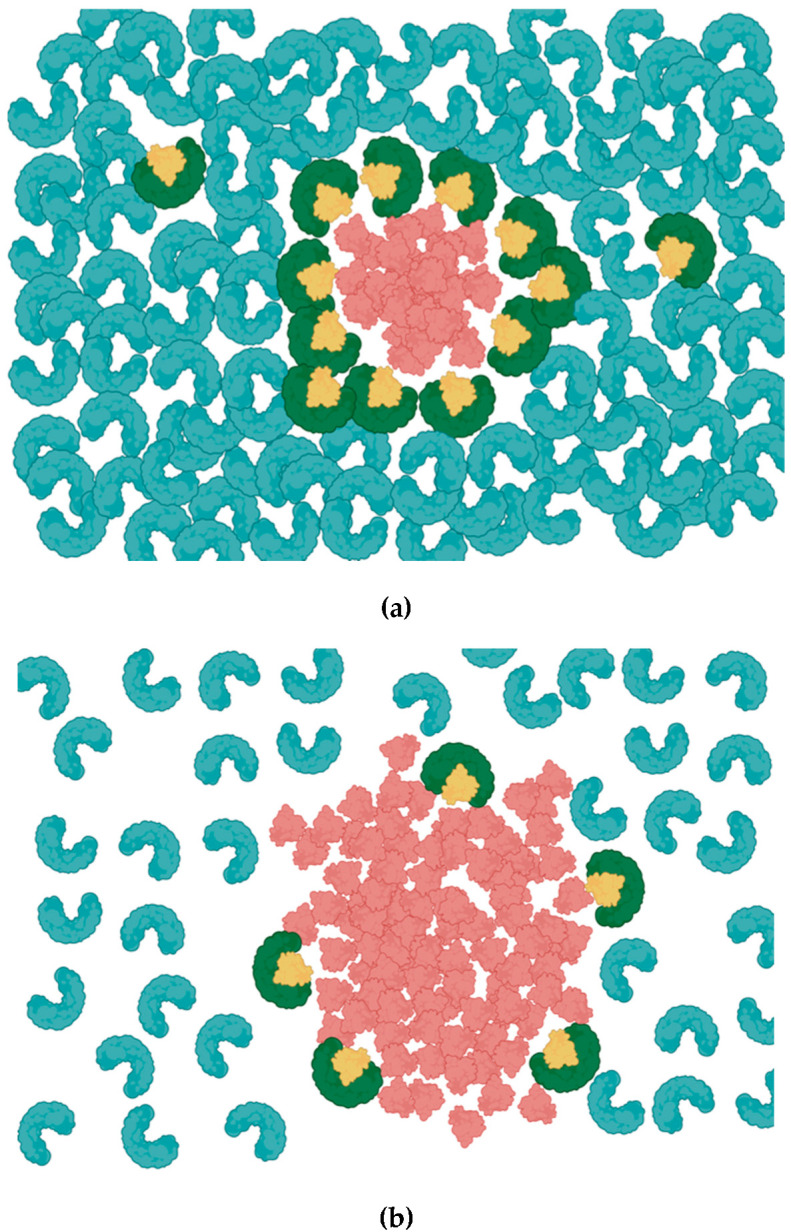


## 2. Neutrophil Serine Protease

Serine proteases are a diverse group of proteolytic enzymes that are characterized by a serine residue in the enzymatic catalytic triad. Serine proteases are widely distributed in the animal and plant kingdoms and homologs are expressed by many viruses [21,22]. Serine proteases are multifunctional and are involved in many biological processes [21]. Among these, the roles of serine proteases in immunity deserve attention because they affect the balance between infections and the immune response. Neutrophils make up approximately 60% of leukocytes and play a prominent role in innate immunity as first-line defenders. Neutrophils carry out important functions in infection by mobilizing from the bone marrow to the circulation and migrating to the site of the infection or injury [23]. Neutrophils employ a range of mechanisms to kill microorganisms such as phagocytosis, secretion of antimicrobial enzymes and release of neutrophil NETs [24]. During their early differentiation in the bone marrow, neutrophils produce several neutrophil serine proteases (NSPs), including NE, cathepsin G (CG), proteinase-3 (PR3), and NSP-4, which are stored in azurophil granules [25,26]. Azurophil granules rapidly fuse with the neutrophil membrane to secrete these proteases, as well as myeloperoxidase (MPO) and defensins, [25,27]. One major function of NPS is to break down ECM, a supportive structure made of two distinct compartments, namely basement membrane and interstitial membrane. These layers are composed of a complex network of proteins such as collagen, elastin and laminin and polysaccharide molecules such as proteoglycans [28]. NE digests elastin, collagen IV, transmembrane proteins, lung surfactant factors and severs tight junctions between cells [28], initiating the vicious cycle of neutrophilic inflammation and declining lung function. Thus, NE has received the biggest share of attention among other serine proteases. 

Furthermore, NE has additional roles in the regulation of inflammatory responses such as the processing and modifying the function of cytokines and chemokines, the inactivation of progranulin, which is an anti-inflammatory mediator, and activation of specific cell surface receptors (e.g., formyl peptide receptors and TLRs) [25,29]. However, exaggerated and prolonged NE activity, unopposed by inhibitors, can lead to tissue damage, vascular injury, cancer and severe cardiopulmonary diseases such as COPD and cystic fibrosis [27,30]. Excessive NE release positively correlates with severe airway obstruction in people with COPD [31], lung injury in severe pneumonia [32], compromised lung permeability and increased inflammatory cytokine secretion that cause ALI and ARDS [33]. On the other hand, depending on the specific microbial defense mechanisms, there are instances, where NE may not provide effective antimicrobial effects, but damage the host tissues. Sahoo et al. [34] demonstrated that, when wild-type mice were infected with *Burkholderia pseudomallei*, they were more susceptible to infection and suffered more lung damage and vascular leakage than mice deficient in NE. Moreover, unmitigated NE is implicated in lung injury by upregulating pro-inflammatory cytokines such as IL-8, promoting airway remodeling and impairing innate immunity. This point is reviewed in [35].

The proteolytic activity of serine proteases is finely regulated by several inhibitory proteins including serpins (serine protease inhibitors), AAT [36], and secretory leukocyte protease inhibitors [37]. AAT has a 25-fold higher association rate constant for NE binding compared to other NSPs, thus NE is the preferred target for AAT inhibition [38,39]. Serpins inhibit protease activity at inflammation sites. AAT and α1-antichymotrypsin (ACT) are the two major serpins found in plasma that inhibit NSPs [40]. Despite abundant information on AAT activity and its function, further studies will be required to understand the full potential of this molecule and its inhibitory mechanisms. 

## 3. Human Alpha-1 Antitrypsin 

AAT is secreted as a 52kD glycoprotein with a half-life of 4–5 days in plasma [41]. It is primarily synthesized and secreted from the liver. However, AAT is also expressed in neutrophils, macrophages, pulmonary alveolar cells, intestinal epithelia, and corneal cells [42]. Humans produce approximately 34 mg/kg/day, which, together with a serum half-life of 4–5 days, accounts for the 1–2 g/L concentration in blood plasma [41]. The broadly acknowledged function of AAT is to protect lung parenchyma from destructive proteolytic enzymes such as NE that can cause irreversible damage to the lungs. Less acknowledged, AAT is a key modulator of many other innate immune functions. Notably, AAT regulates the expression of CD14, a major receptor on the monocyte surface that drives inflammation in response to bacterial lipopolysaccharide (LPS) [43], inhibits tumor necrosis factor–α (TNF-α) expression and signaling in neutrophils both in vitro and in vivo [44], regulates neutrophil chemotaxis, inhibits activation of human monocytes [45], and reduces viral effects against HIV-1 via inhibiting nuclear factor-κB (NF-κB) [46], and SARS-CoV-2 by inhibiting viral entry [47]. Intriguingly, AAT is also being considered as a potential therapeutic candidate for type 1 diabetes. Gp96 and HSP70 are heat shock proteins released from the damaged β-islet cells and act as damage-associated molecular patterns (DAMP) to induce inflammation. AAT can bind and inhibit these proteins and prevent further β-islet cell injury and provides protection against diabetes [48]. Yet, the mechanism of binding between AAT and these two molecules is unknown and needs further work [49]. 

## 4. Alpha-1 Antitrypsin Structure and Inhibitory Mechanism

AAT consists of 394 amino acids, including one free cysteine residue at position 232 and asparagine residues that serve for N-glycan attachments at positions 46, 83 and 247 [39,41]. AAT occurs in several isoforms that are distinguished by posttranslational modifications (PTM). In response to inflammation, circulating AAT diffuses into the lung and inhibits elastase activity [39]. The inhibition mechanism of AAT is different from a typical lock and key mechanism. AAT undergoes an unusual conformational change following cleavage by the protease and transitions from a high energy to a stable low energy state before the acyl-enzyme intermediate is hydrolyzed. AAT’s structure consists of three β-sheets and 8–9 alpha helices. Serpins including AAT are “single-use” suicidal inhibitors that cause irreversible inhibition of proteases. The inhibition mechanism includes cleavage of the reactive center loop (RCL) of AAT and permanently alters the active site and the function of the protease. AAT and the protease form a 1:1 complex (Figure 2). To form the complex, AAT undergoes a large conformational change and physically traps the protease. This is an intermediate step in the enzymatic reaction and can be stable for weeks in vitro. As result, both the enzyme and the inhibitor are covalently linked [50]. 

## 5. Alpha-1 Antitrypsin Deficiency

AAT protein is encoded by the SERPINA-1 gene located on human chromosome 14, for which more than 100 mutations have been described and additional genetic variants are likely to be discovered [51,52]. The wild-type allele coding for the normal protein is noted as M (M1–M6; serum level ≥ 0.9 mg/mL), whereas most of the AAT variants are in S and Z forms comprising three major genotypes: ZZ, SZ and SS that are associated with reduced serum levels compared to wild-type. The homozygous (ZZ) genotype in the European population is estimated at approximately 0.01–0.02% [53,54] A study carried out with 224 cohorts across 65 countries estimated that 253,404 individuals possess the ZZ genotype worldwide. The Z mutation is located at the base of the RCL loop and affects the loop structure and causes severe AAT deficiency [55].

The AAT deficiency (AATD) is a common, yet underappreciated hereditary disorder, which is characterized by insufficient levels of circulating AAT, and a high neutrophil burden (neutrophilia), both of which predispose toward hepatic and pulmonary diseases. People with AATD are at high risk of emphysema and COPD. In fact, AATD is the only genetic factor that is proven to be responsible for COPD [56]. The most prominent consequence of AATD is the development of severe lung emphysema, as the lung fails to maintain a protease-antiprotease balance, which is crucial for controlling excessive proteolytic damage caused by neutrophil serine proteases (NSP) [57]. To date, augmentation therapy has been the most recognized method to treat severe emphysema caused by AATD. However, AAT augmentation can only slow down the disease progression to a certain degree, and it can only delay lung failure. Moreover, frequent blood infusions are inconvenient for the patient. Therefore, many efforts are being made to produce a recombinant AAT protein with modifications to improve the yield, specificity, and half-life. Recombinant AAT can not only be used as replacement therapy for AAT deficiency, but it may find applications in other disease conditions [58]. 

Individuals with AATD have exaggerated neutrophilic activity in their lungs compared to healthy individuals. Elastase released by neutrophils triggers alveolar macrophages to produce leukotriene B4 (LTB4), which acts as a chemoattractant for neutrophils [59]. O’Dwyer et al. [60] demonstrated that AAT can form a complex with LTB4, thus modulating BLT1 (Leukotriene B4 receptor 1) binding and inhibiting downstream signaling pathways. Consequently, lack of AAT may lead to excessive neutrophil infiltration. In addition, the absence of AAT allows the excessive migration of neutrophils toward IL-8. Bergin et al. [61] demonstrated distinct mechanisms of how AAT prevents neutrophil chemotaxis in response to IL-8 and soluble immune complexes (sIC). AAT forms a complex with IL-8 and as a result, AAT inhibits IL-8 binding to its receptor CXCR1. This inhibits downstream signaling, including AKT-phosphorylation, calcium flux, and cytoskeleton rearrangements, subsequently leading to reduced neutrophil infiltration. The anti-chemotaxis effects of AAT via sIC engagement with ADAM-17 (A disintegrin and metalloprotease 17) activity hampers the release of FcγRIIIb and affects downstream signaling [61]. The authors also reported the colocalization of AAT and FcγRIIIb in the neutrophil membrane by coimmunoprecipitation. ADAM-17 activity itself is also elevated in AATD individuals, due to the increased TNF-α release from neutrophils. These studies have highlighted the anti-inflammatory mechanisms of augmentation therapy. Upon administration of AAT in ZZ-AATD individuals, elevated levels of AAT in the serum bound to IL-8, FcγRIIIb and LTB4, thus in concert with downregulating excessive neutrophil chemotaxis. However, circulating blood contains approximately 1–2 g/L of AAT under normal conditions, a level that is difficult to consistently maintain with the administration of AAT. 

Although COPD and liver disease are widely recognized as consequences of inadequate or aberrant AAT, there are other potential consequences of AATD, including autoimmune diseases, type 1 and type 2 diabetes, and obesity. Mansuy-Aubert et al. [62] demonstrated that the NE/AAT ratio can be considered an important molecular marker of obesity. They reported that deficiency of leptin, a hormone regulating satiety is positively correlated with reduced AAT and increased NE levels in serum in high-fat diet (HFD) induced obese mice. In these mice, leptin stimulates AAT gene expression through the Jak2-Stat3 pathway. These results indicate the importance of AAT as a regulator of obesity-related inflammation. Severe cases of AATD are also associated with skin panniculitis whose hallmark is inflammation of the subcutis [63]. Symptoms of panniculitis include red nodules, painful lesions, and discharge. Some estimates put the prevalence of the panniculitis in AATD population at less than 1%, but this is likely an underestimate because AATD is underdiagnosed in patients with panniculitis due to a lack of awareness about AATD as a contributing factor [64]. 

During the SARS-CoV-2 pandemic, studies were carried out in search of an association between disease severity and AATD. The efforts produced mixed results due to low numbers of cohort participants in some studies. Conversely, many reported that patients with severe AATD have an increased risk of COVID-19 infection. Despite its association with lung and liver pathology, AATD is largely overlooked and underdiagnosed. It is recommended that every patient with COPD be screened for AATD to facilitate the early diagnosis of AATD, distinguish them from other COPD patients, and provide suitable treatment early on. Individuals with both COPD and AATD should also consider AAT therapy besides normal COPD treatment. Intravenous AATD augmentation therapy is the currently accepted treatment method, and it has shown promise in attenuating the loss of lung function [53]. Given the burden on the overall health of individuals with severe AATD, despite being an expensive approach, AAT augmentation therapy is safe to establish circulating AAT levels above the protective threshold. Production of recombinant AAT with a greater half-life and application of gene therapy might provide more versatile and comparatively inexpensive treatment in future.

## 6. Effects of NETosis in Respiratory Diseases and COVID-19 

NETosis is a programmed cell death process that was identified as an antimicrobial defense mechanism against invading bacterial, fungal viral pathogens [5]. Neutrophils expel their citrullinated chromatin into the extracellular environment. These DNA structures, called NETs are decorated with proteins from granules and cytoplasm [65,66]. Peptidyl arginine deiminase type 4 (PAD4), NE and gasdermin D (GSDMD) play a vital role in NET formation [67]. NE activates GSDMD by cleaving at several positions between 275 and 282. GSDMD in turn facilitates NE translocation from granules into the cytoplasm by forming pores in the granule membrane [8]. Proinflammatory cytokines such as IL-8 and IL-1β and activation of pattern recognition receptors (PRR) upregulate calcium mobilization and ROS production, which then instigate the activation and translocation of PAD4 and NE into the nucleus. PAD4 converts peptidyl-arginine into peptidyl- citrulline and NE cleaves histone proteins, causing decondensation of chromatin [68,69]. GSDMD generates pores in the plasma membrane, thus releasing chromatin fibers and granule proteins from the cell [8]. 

Besides trapping microbial pathogens, NETs also bind to red blood cells and platelets causing coagulation and may act as a significant contributor to microvascular thrombosis [70]. Persistent release and inefficient degradation of NETs contribute to the severity of many viral diseases such as community-acquired pneumonia (CAP), COVID-19 [11,67,71], acute and chronic airway disorders [72,73,74], cancer metastasis and organ damage [67,75,76]. NETs may have implications in the pathogenesis of autoimmune diseases such as systemic lupus erythematosus (SLE) due to being a source of autoantigens [77]. Hence, effective management of NETosis has become a potential therapeutic intervention in numerous diseases. However, understanding the role of aberrant NET production in pulmonary diseases and its underlying regulatory mechanisms remains incomplete. Neutrophils have similar protective mechanisms and deleterious effects on any viral respiratory infection including COVID-19, influenza pneumonitis [78], and RSV [79]. Thus, progressive lung impairment due to NETopathy-related complications is common among all these infections and neutrophils have a specific protective role in respiratory infections. However, excessive viral loads induce severe pathology that reflects, in large measure, the unfavorable outcomes of neutrophils. 

Lefrançais et al. [73] have demonstrated that treatment with DNase I together with vancomycin in mice infected with MRSA significantly reduces mortalities and ameliorates lung damage and lung vascular permeability. Patients with infection related ARDS had augmented NET remnants in plasma, which was associated with poor outcomes such as high disease severity and mortality [80]. This observation was confirmed in clinical studies including CAP [71] and COVID-19 [76,81,82,83]. Incubation of neutrophils from healthy donors for 6 h with 10% of plasma obtained from hospitalized patients with COVID-19, significantly elevated NETs release, indicating that plasma from COVID-19 patients can induce NETosis. However, plasma factors that induce NETs are not yet identified. One speculation is that NETs concentration in plasma is elevated, due to an increase in the number of circulating neutrophils [84] rather than virus-induced effects. However, accumulated evidence shows that neutrophils from patients infected with SARS-CoV-2 had an increased ability to produce NETs [82,85] largely via PAD4 dependent pathway [76,82], with increased ROS production mediated by SARS-CoV-2, as verified by Arcanjo et al. [86] NET release has also been demonstrated to have multiple negative outcomes in COVID-19. Approximately, 10–15% of the COVID-19 patients manifest ARDS, whose severity was correlated with the magnitude of the cytokine storm, increased vascular permeability, and microvascular thrombus formation mediated by higher neutrophil infiltration and increased NET formation [87]. Overproduction of pro-inflammatory cytokines such as TNF-α, IL-6, IL-8, IL-10 and IL-17, and endothelial dysfunction caused by the virus are therefore major concerns in COVID-19 associated ARDS [20,88]. 

Scientific investigations put the spotlight on the interplay between SARS-CoV-2, microthrombosis and NETosis in as they may hold the key to understanding disease severity. NETs activate platelets and contribute to their aggregation, thus promoting the coagulation cascade in thrombosis that is observed during disease progression and organ failure in COVID-19 patients [67,89,90,91]. In fact, 91.3% of deaths due to COVID-19 show evidence of microthromboses, thus highlighting the importance of therapies that target thrombosis [92]. Immunofluorescence staining of postmortem lung specimens of COVID-19 patients clearly demonstrated the presence of NETs and thrombi in the areas of inflammation causes obstruction of alveoli and bronchioles and leads to lung fibrosis [93]. Analysis of 82 COVID-19 infected individuals demonstrated that patients with severe disease had higher levels of NET formation, which correlated with elevated amounts of NE, IgA2 antibody and extracellular DNA in sera [94]. It was further postulated that IgA2 antibody induces NETosis in SARS-CoV-2 infection [94]. A link between IgA2 levels and NET formation was also reported by Steffen et al. [95] Additionally, citrullinated histone H3 was elevated in the serum of COVID-19 patients and correlated with platelet activation [83]. Taken together, the NET formation can be considered a potential biomarker of COVID-19 disease severity. Therefore, devising new drugs to target NETosis directly or indirectly could show potential to reduce tissue damage and neutrophil-mediated inflammation in COVID-19 and other lung diseases. Although many authors have underscored the role of NETs and microthrombosis in COVID-19, it is a complicated association that required more randomized trials to provide evidence-based reasoning. 

## 7. AAT Supplementation to Reduce NET-Mediated Pathogenesis 

An increasing number of drug studies have been carried out targeting NE, PAD4 and GSDMD, as means of inhibiting NETosis [76]. NE mediates NETosis by disassembly of the cytoskeleton (actin and vimentin) and induces chromatin decondensation independent of PAD4 [96]. Neutrophils from NE knockout mice or neutrophils treated with NE inhibitors show reduced NETosis in response to PMA, or C. albicans [7,97]. An increase in NET release is observed in neutrophils that are deficient in protease inhibitor (serpinb1−/−), when stimulated with NET activating agents such as PMA, LPS (*P. aeruginosa*), or platelet-activating factor (PAF). Further, treatment of neutrophils with recombinant protease inhibitor (rSerpin B1) decreased NET formation induced by PMA, LPS and PAF stimulation [97]. It was demonstrated that SARS-CoV-2 induces NETosis via the GSDMD-dependent pathway, which requires the entry of the virus via ACE2 or serine protease TMPRSS2 and viral replication [76]. NETosis was abrogated in the absence of viral entry or if the viral replication was inhibited. Together, these data show the potential of AAT as a therapeutic, which could regulate NETosis and neutrophil hyperinflammation. 

In a unique study analyzing the effect of AAT on NETs, Frenzel et al. [98] compared neutrophils from ZZ, and AATD patients before and after AAT augmentation therapy. This study revealed that supplementation with AAT does not inhibit the formation of NETs, but markedly changes the structure of NETs and increases the size of neutrophil aggregates. In a subsequent experiment, the authors compared NET formation from healthy donors and from ZZ ATTD augmented patients and observed differences in NET structure, containing disrupted chromatin fibers in ZZ AATD patients, whereas healthy donors produced NETs of the expected structure. Yet, the differences between test and control NETs observed in above study were subtle and difficult to compare. 

A separate study suggesting that natural differences in protease inhibitors affect NET formation reported that human newborns have tightly regulated mechanisms to restrict NET formation [99]. Reduced capacity for NET formation may have physiological significance, as increased exposure to inflammatory stimuli may cause direct tissue injury as well as other short-term or long-term damage to the immune system. A small protein of around 4 kDa with an identical sequence to 18 amino acids in the carboxy terminus of AAT, named neonatal NET Inhibitory factor (nNIF) was isolated from umbilical cord blood. Further, umbilical cord blood is abundant with AATm^358^ (carboxy terminus fragment with 44 amino acids; previously described in Niemann et al. [100], which will be cleaved at the carboxy terminus presumably by protease 1. nNIF inhibits the major events of NETosis including PAD4 activation, histone citrullination, and nuclear decondensation. Yost et al. [99] argue that nNIF can inhibit NET formation, a hypothesis that is consistent with the data from Frenzel et al. [98]. Thus, AAT, or its proteolytic fragments, may have potential therapeutic value in patients who are at risk of developing excessive NETs and may prevent lethal consequences such as ischemic heart attacks. Middleton et al. [81] have demonstrated that synthesized nNIF can reduce NET formation in neutrophils isolated from the plasma of COVID-19 patients. 

In summary, AAT supplementation cannot completely eradicate NET formation. However, in COVID-19, it was discovered that neutrophils tend to produce more NETs when infected with the virus. Thus, we speculate neutrophil-derived AAT might have an important role in regulating this phenomenon. 

## 8. Role of Neutrophil-Derived AAT

Hepatocytes are the major source of circulating AAT in the blood. However, neutrophils also produce AAT and store the protein in azurophil granules along with NE and both are secreted following stimulation. Clemmensen et al. [101] studied the localization of AAT in neutrophils and determined that neutrophils produce AAT during maturation and enhance its production after entering into the blood circulation and following migration into tissues [101]. It is intriguing to consider whether neutrophil-derived AAT makes a significant contribution to NE inhibition, what prevents the direct inhibition of NE in the azurophil granules and how complex formation is prevented. If AAT and NE are stored in the same cellular compartment and released at the same time, there presumably is a mechanism to regulate one or both protein functions. Previous studies demonstrated that neutrophils store serine proteases, NE, PR3 and CG in azurophilic granules in their active form [25], but AAT does not form complexes with those serine proteases in the granules [102,103]. Therefore, we could assume that AAT is in its inactive form in the azurophil granule. Conceivably, AAT gains inhibitory activity following additional modifications that respond to specific stimuli. However, further studies are needed to establish the mechanism of how AAT is partitioned with proteases in the granule, and how it is secreted and activated. 

The general perception is neutrophils produce a relatively small volume of AAT compared to hepatocytes (https://genome.ucsc.edu/index.html, accessed on 5 June 2022). However, it is not quite accurate to compare these two sources of AAT because the liver is the major source of AAT and hepatocytes continually produce AAT in relatively large quantities regardless of inflammation. On the contrary, neutrophils secrete AAT locally and only when there is a stimulus. Combined with the large accumulation of neutrophils at sites of inflammation that leads to neutrophil swarms could raise the AAT concentration in a focused manner far beyond the levels that are achieved by distantly produced, hepatocyte-derived AAT.

## 9. AAT Protection against COVID-19

Since the outburst of COVID-19, more than five hundred studies were performed to identify the antiviral and anti-inflammatory response of AAT to SARS-CoV-2 and how AATD contributes to disease severity. According to the Alpha-1 foundation, 32% of SARS-CoV-2 infected AATD patients have been either hospitalized or needed to have critical care following the infection by COVID-19. However, the AATD population is not specifically considered in vaccination trials to assess the efficacy and effectiveness of COVID-19 vaccines [104]. Wettstein et al. [47] analyzed their previously generated peptide/protein libraries derived from bronchoalveolar lavage for inhibitors of SARS-CoV-2 spike-driven entry. They identified AAT as a highly abundant major antiviral peptide in circulation (0.9–2 g/L, 17–38 µM) that blocks SARS-CoV-2 at the entry point. They further confirmed the rationale for this observation with in vitro studies that AAT inhibits TMPRSS2 in a dose-dependent fashion and is effective at physiologically relevant concentrations of 5–50 µM. TMPRSS2 is an essential type II transmembrane serine protease expressed in the lungs [105]. It cleaves the spike protein of SARS-CoV-2 for viral intracellular entry. A recent study with a cohort of 40 COVID-19 patients demonstrated that acute SARS-CoV-2 infection correlated with significantly increased blood AAT levels, and it may serve as a natural suppressor of novel coronavirus replication [106]. In [107] and others have elucidated that AAT inhibits TMPRSS2 in a similar fashion to protease AAT inhibitory mechanism and its inhibitory activity was comparable with camostat mesylate, a drug that inhibits TMPRSS2, which is currently under investigation in several clinical trials to treat COVID-19 [108]. Conversely, an in vitro study has shown that AAT inhibitory activity on SARS CoV-2 replication was only moderately effective compared to camostat. Overall, these studies propose that AAT has the potential to limit SARS-CoV-2 viral load in host cells by modulating TMPRSS2. 

Additionally, AAT has also been shown to have inhibitory effects against disintegrin and metalloproteinase 17 (ADAM17), which mediates ACE2, a SARS-CoV-2 entry receptor [109]. Belouzard et al. [110] showed the important role of elastase-mediated SARS-CoV fusion with the host cell membrane via cleavage of spike protein at S2 position in vitro. However, NE has attracted only a little attention in this perspective to date due to its less specific cleavage at virus spike protein in SARS-CoV-2. On the contrary, in a recent publication in vitro, it was revealed that both NE and PR3 engage in priming virus S protein by cleaving close to the polybasic sequence at S protein interface [17]. However, SARS-CoV-2 activation by in vivo sources of elastase is not been experimentally tested to date. NE upregulates tissue factor (TF) activity and coagulation by degrading tissue factor pathway inhibitor (TFPI), an endogenous anticoagulant protein leading to arterial thrombosis in infections in vivo [111]. AAT drugs might constitute a potential treatment for COVID-19 and this can be a potential target of antiviral intervention. AAT augmentation therapy has been used for decades to treat emphysema in AATD patients [53]. Intravenous infusion of AAT has been monthly administered for 12 months at the rate of 250 mg/kg increased serum AAT concentration by 5-fold in lung epithelial lining fluid without causing side effects [112]. A recently published pilot study [113] has reported that intravenous administration of AAT at the rate of 120 mg/kg/wk is even safe to use and effective compared to the current standard single dose (60 mg/kg/wk) treating AATD. They have reported a marked reduction in inflammatory biomarkers in BALF that are responsible for leukocyte activation, chemotaxis, and recruitment, reduction in NE and CG activity and elastolysis markers in BALF. In vitro transcriptomic data derived from 3D lung model, people who are less susceptible to COVID-19 express higher levels of antiprotease genes including SERPINA-1 [114]. Growing evidence suggests that AAT can be a suitable candidate to treat COVID-19 due to its anti-viral and anti-inflammatory roles.

However, the clinical effectiveness of these doses against SARS-CoV-2 is yet to be confirmed. Recent studies have reported that AAT provides less protection against acute inflammation responses to COVID-19. Despite having elevated AAT levels in blood plasma as the disease progressed into the acute phase, it was not sufficient to overcome the effects of increased IL-6 burden [47,106]. A cohort of 20 people with severe illness who required ICU admission compared to people who had been hospitalized but were stable and patients with community-acquired pneumonia (CAP), had significantly higher IL-6/AAT ratio. The authors also observed the elevation of AAT sialylation, which was previously demonstrated to improve the effectiveness of the anti-inflammatory response. A high IL-6/AAT ratio was associated with prolonged ICU stays followed by a higher incidence of mortality. It appears likely that AAT augmentation therapy with a combination of anti-cytokine therapy would be more beneficial in treating severe COVID-19 infections. 

## 10. Conclusions

To conclude, AAT has been identified as a multifunctional protein produced mainly by the liver in large quantities to safeguard tissue damage from excessive activity of NE. The emergence of COVID-19 has drawn lots of attention to AAT for its antiviral and anti-inflammatory roles. Further, individuals with AATD were identified as a risk group in COVID-19 for high morbidity and mortality. Moreover, the ability of AATD patients to mount a quick immune response to COVID-19 vaccines must be investigated. Given the fact that AAT regulates viral entry, NETosis and thrombosis, AAT augmentation therapy was recognized as a potential treatment to improve the disease outcomes of COVID-19. 

## Figures and Tables

**Figure 2 biomolecules-13-00082-f002:**
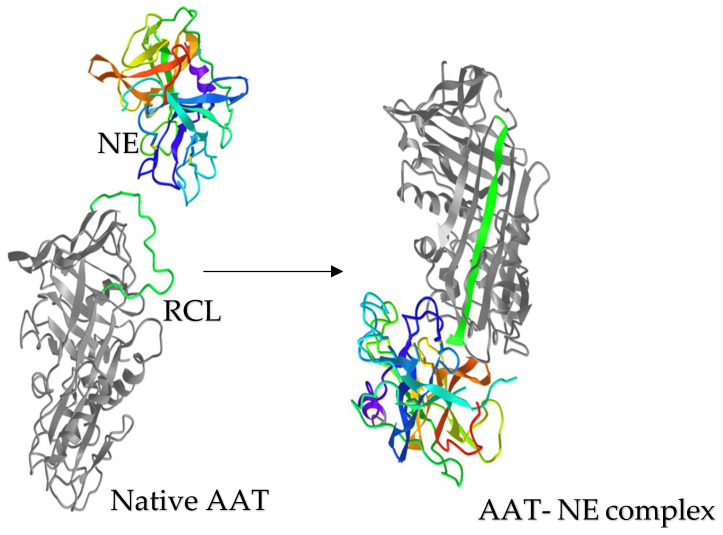
AAT- NE complex formation. Neutrophil elastase (NE) and AAT, shown at the same relative scale. Upon proteolytic cleavage by NE, the reactive center loop (RCL;green) of AAT is incorporated to form the central b strand of the A-sheet (green, right). This conformational change translocates NE from its original position and inactivates AAT activity, while trapping NE till the complex dissociation. Structures are from the Protein Data Bank under accession numbers, AAT: 1ATU; NE: 3Q76 and AAT-NE complex: 2D26.

## Data Availability

Not applicable.

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
