# Peer review of "COVID-19 Pathology Sheds Further Light on Balance between Neutrophil Proteases and Their Inhibitors"

_biomolecules, 2022, doi:10.3390/biom13010082_

Round 1

Reviewer 1 Report

The manuscript by Vasuki Silva and Marko Radic reads well, and the authors discussed the topics properly. Here are my suggestions to improve the manuscript:

-The authors must prepare a graphical abstract to visualize their concepts better.

-What is specific in COVID-19 patients vs. other patients with other viral respiratory infections that make neutrophils so distinct in the pathogenicity of the disease? Please discuss.

-The authors must critically comment on every issue they discussed and clearly express their points of view.

-The authors need to include a paragraph(s) to summarize the discussions and provide the readers with recommendations for future works.

Author Response

Dear Reviewer,

Thank you for your valuable comments and suggestions

Following is the response to each point:

  1. The authors must prepare a graphical abstract to visualize their concepts better. -Attached as a PDF document
  2. What is specific in COVID-19 patients vs. other patients with other viral respiratory infections that make neutrophils so distinct in the pathogenicity of the disease? Please discuss.

To our knowledge, neutrophils have similar protective mechanisms and deleterious effects on any viral respiratory infection including COVID-19, influenza pneumonitis (Narasaraju etal), RSV (Muraro etal) etc. For example progressive lung impairment due to NETopathy related complications is common among all these infections. Therefore, neutrophils have a specific protective role in respiratory infections. But excessive viral loads induce severe pathology that reflects, in large measure, the unfavorable outcomes of neutrophils. 

References

Narasaraju T, Yang E, Samy RP, et al. Excessive neutrophils and neutrophil extracellular traps contribute to acute lung injury of influenza pneumonitis. Am J Pathol. 2011;179(1):199-210. doi:10.1016/j.ajpath.2011.03.013

Muraro SP, De Souza GF, Gallo SW, et al. Respiratory Syncytial Virus induces the classical ROS-dependent NETosis through PAD-4 and necroptosis pathways activation. Sci Rep. 2018;8(1):14166. Published 2018 Sep 21. doi:10.1038/s41598-018-32576-y

3. The authors must critically comment on every issue they discussed and clearly express their points of view.- added to the manuscript

4. The authors need to include a paragraph(s) to summarize the discussions and provide the readers with recommendations for future works.- Summary paragraph added

Thank you again for your time

Reviewer 2 Report

Really nice written review.

I think the title is a bit misleading to the extent of actually discussing NE and COVID in the text. So maybe the title could be adapted?

NSPs? Neutrophil serine proteases? I did not find the full name, only abbreviation.

IS the concentration of NE in plasma know?

How high is the incidence of AATD?

How is ATT lost in C19 patients with ADRS? line 88

Check sentences in lines 384-387.

Author Response

Dear Reviewer,

Thank you very much for your comments and suggestions

Following is the response:

  1. I think the title is a bit misleading to the extent of actually discussing NE and COVID in the text. So maybe the title could be adapted?

Title is changed. 

2. NSPs? Neutrophil serine proteases? I did not find the full name, only abbreviation.- Added

3. IS the concentration of NE in plasma know? Plasma NE concentration should be very low due to abrupt neutralization of AAT, unless there is a AAT deficiency

4. How high is the incidence of AATD? According to Alpha-1 foundation there is nearly 100,000 with ZZ deficiency in US. In Europe its 0.01-002%

5. How is ATT lost in C19 patients with ADRS? line 88

Due to high NE concentration, AAT is either complexed or cleaved. This leads to the imbalance

5. Check sentences in lines 384-387.- Fixed

Thank you again for your time

Round 2

Reviewer 1 Report

The GA needs a legend associated with it.

Please add your explanation regarding my question about what is specific in COVID-19 patients vs. other patients with other viral respiratory infections that make neutrophils so distinct in the pathogenicity of the disease? as a paragraph into the Discussion.

Author Response

Dear reviewer,

Thanks for the comment. Paragraph is now added : 319-325 

For the GA., following legend is added:

Alpha-1 Antitrypsin (AAT; blue) inhibits neutrophil elastase (NE; pink) in a healthy individual (left). AAT and NE make a stable 1:1 complex (AAT; green, NE; yellow), which limits the NE spread. Under conditions of reduced AAT, as in AAT deficiency (AATD), NE is inadequately inhibited, spreads unopposed, and may damage healthy tissues. In SARS-CoV-2, AAT reduces viral entry, NETosis, and microthrombosis. Individuals who lack AAT protection, therefore, are at higher risk of morbidity and mortality.

Best Regards

Authors
